# Exploring the Trade-off between Quality and Diversity of Language Models during Reinforcement Learning

## Abstract

Reinforcement learning (RL) has become the dominant approach for post-training autoregressive language models, but a recurring challenge is that improvements in *quality* often come at the expense of *diversity*, which is a practical concern in exploratory domains such as scientific discovery. Although this trade-off is widely acknowledged, it has lacked a quantitative characterization. In this work, we systematically investigate the quality-diversity dynamics of RL finetuning of language models primarily on molecular generation, a domain where diversity is both essential for discovery and quantitatively measurable.

Across RL checkpoints, we observe that mean quality ($\mathcal{R}$) and diversity ($\mathcal{D}$) trace a smooth trajectory captured by a robust exponential law, $\mathcal{R} = -a \cdot \exp(c \cdot \mathcal{D}) + b$, independent of step indexing. Extending prior work on quality-entropy trade-offs, we further show that quality also follows an exponential relation with sampling entropy ($\mathcal{H}$), $\mathcal{R} = -a_0 \cdot \exp(c_0 \cdot \mathcal{H}) + b_0$, with $c_0$ quantifying exploratory progress. An approximately linear link between entropy and diversity explains why the two laws compose, and an information-theoretic illustration clarifies the role of the exponential form. We also conduct ablations on influencing factors, including model scaling, reward shaping, and training setup, validate these findings across multiple generation objectives, and extend the experiments to textual exploratory creativity tasks by large language models. Finally, we demonstrate how the fitted laws provide actionable guidance for RL finetuning of language models on exploratory tasks. Overall, our study moves beyond qualitative accounts of diversity collapse, offering a compact quantitative model, an underlying entropy-based mechanism, and practical tools for exploratory RL with language models.

## 1 Introduction

Reinforcement learning (RL) has become a central paradigm for post-training autoregressive language models, enabling policies that better align with downstream objectives ranging from factuality and safety to domain-specific preferences (Ouyang et al., 2022; Lee et al., 2023; Guo et al., 2025; Zhang et al., 2025a). A recurring observation, however, is that optimizing for *quality* (task reward or accuracy) typically sharpens the model's output distribution, thereby reducing *diversity* (coverage of plausible alternatives) (Zhang et al., 2020; Sun et al., 2025; Li et al., 2025a). In creative and exploratory domains such as brainstorming, creative writing, and scientific discovery, this tension is consequential: a language model that attains high average quality but collapses onto a narrow portion of the solution space can miss novel, high-value candidates (Singh & Pelaez, 2008; Zhang et al., 2025b). Despite widespread acknowledgment of this phenomenon, the relationship is usually described qualitatively as a "trade-off," with little consensus on its quantitative form.

Recent works have begun to quantify adjacent facets of this problem. Cui et al. (2025) investigates the interplay between RL reward and sampling entropy for reasoning language models, reporting an exponential law between reward and entropy. Yet, reasoning benchmarks primarily value correctness and are largely indifferent to sample diversity; as a result, those findings do not directly establish a quantitative link between quality and diversity. More broadly, quality-diversity algorithms in evolutionary computation formalize the goal of producing diverse, high-performing sets, but they rely

on hand-crafted behavioral descriptors that do not readily transfer to language modeling (Lehman & Stanley, 2011; Mouret & Clune, 2015).

**Our perspective.** We argue that a rigorous understanding of quality-diversity trade-off in reinforcement learning of language models should (i) measure both axes with domain-appropriate, quantitative metrics; (ii) characterize the trajectory a policy follows during RL finetuning; and (iii) explain the mechanism that drives the resulting curve. To that end, we ground our study mainly in molecular generation, a scientific-discovery task where output diversity is intrinsically valuable and, crucially, quantifiable in chemical space (Du et al., 2022). Compared to natural language, this setting offers: (1) a direct practical need for diverse outputs to enable exploration; (2) a clear, standardized diversity metric; and (3) an evaluation regime where the maximal quality $\mathcal{R}_{\max}$ is attainable under RL, allowing us to isolate and analyze the quality-diversity trade-off rather than conflate it with unattained quality ceilings.

**A quantitative law along the RL trajectory.** Monitoring the policy across checkpoints during RL finetuning, we find that the points of mean quality $\mathcal{R}$ and diversity $\mathcal{D}$ (each computed from fresh samples at every checkpoint) concentrate on a smooth trajectory that is accurately captured by an exponential form:

$$\mathcal{R} = -a \cdot \exp(c \cdot \mathcal{D}) + b, \qquad a, b, c > 0, \quad (1)$$

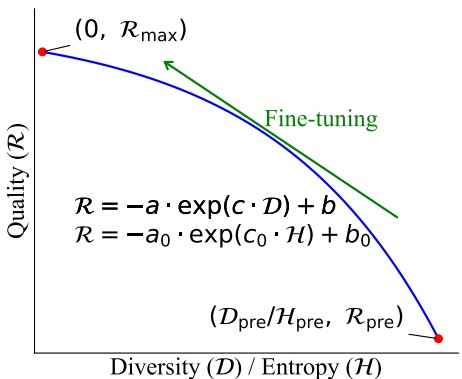

traversed from lower-right (low $\mathcal{R}$, high $\mathcal{D}$) to upper-left (high $\mathcal{R}$, low $\mathcal{D}$) as training progresses, as shown in Figure 1. Importantly, the curve in Equation 1 is irrelevant to the indexing of steps: it characterizes the attainable quality-diversity trade-off for the task, the pretrained model ($\mathcal{R}_{\mathrm{pre}}$, $\mathcal{D}_{\mathrm{pre}}$), and the quality ceiling $\mathcal{R}_{\max}$.

Figure 1: Illustration of the quantitative quality-diversity and quality-entropy trade-offs we observe.

**Entropy as the hidden driver.** The empirical law in Equation 1 invites an underlying mechanism. Extending Cui et al. (2025), we show that quality also follows an exponential law with sampling entropy $\mathcal{H}$ during RL finetuning:

$$\mathcal{R} = -a_0 \cdot \exp(c_0 \cdot \mathcal{H}) + b_0, \qquad a_0, b_0, c_0 > 0, \quad (2)$$

with $c_0$ acting as an exploration coefficient. Empirically, $\mathcal{H}$ is approximately linearly related to $\mathcal{D}$ in molecular generation, which explains why Equation 1 emerges when Equation 2 holds. We further provide an information-theoretic illustration; the resulting form naturally yields the exponential relationship and clarifies the role of $c_0$ as the balance between convergence and exploratory alignment.

**Scope and implications.** Beyond a single task, we replicate the analysis across additional generation objectives and ablations of model size, reward transformations, and finetuning setup. Two regularities stand out. First, the quality-diversity trajectory is fundamentally determined by the task, the initial point (i.e., the pretrained model), and the ceiling quality. Second, the fitted law can practically act as a predictor for the expected diversity attainable at a target quality, and vice versa, thereby informing decisions in exploratory RL workflows (Section 3). In addition, we also test whether the quantitative laws persist in non-chemical domains. In particular, we include exploratory creativity tasks (circle construction and line construction (Nagarajan et al., 2025)) using large language models, finding that the same exponential forms hold with high fidelity, which validates the cross-domain generality of the observed phenomenon.

In summary, our paper makes the following contributions:

- **Quantitative law of quality-diversity.** We observe that the quality-diversity trade-off traced by RL checkpoints of language models ~~for molecular generation~~ follows a simple, robust exponential law $\mathcal{R} = -a \cdot \exp(c \cdot \mathcal{D}) + b$ with tight fits.

- **Entropy mechanism.** We establish a companion exponential law between quality and sampling entropy, $\mathcal{R} = -a_0 \cdot \exp(c_0 \cdot \mathcal{H}) + b_0$, and explain why the exponential form arises using an information-theoretic illustration. Essentially, the coefficient $c_0$ quantifies exploratory progress compared to previous work (Cui et al., 2025). The entropy mechanism explains the observed quality-diversity law via an approximately linear relation between entropy and diversity in molecular generation.

- **Task/domain generality and ablations.** We validate the laws across multiple molecular generation objectives (JNK3, GSK3b, QED) and textual creativity tasks (circle and line construction), and analyze influencing factors (model scaling, reward shaping, and training setup), showing that these factors in RL finetuning do not significantly affect the quality-diversity trajectory of language models.

- **Practical guidance.** The fitted laws can be translated into actionable tools for estimating attainable diversity at target quality, and motivating model ensembles to exceed single-policy diversity in exploratory scenarios.

## 2 QUANTIFYING THE QUALITY-DIVERSITY TRADE-OFF

### 2.1 PRELIMINARIES

#### 2.1.1 AUTOREGRESSIVE LANGUAGE MODELS AND RL FINETUNING

We study autoregressive language models finetuned with reinforcement learning. The autoregressive generation process samples tokens iteratively,

$$x_t \sim p_\theta(x_t \mid \boldsymbol{x}_{<t}), \tag{3}$$

where $\theta$ are model parameters and $\boldsymbol{x}_{<t}$ the prefix at step $t$. RL techniques have fundamentally transformed the training paradigm of language models (Ouyang et al., 2022; Zhang et al., 2025a). Some exploratory applications of language models emphasize either *quality* (task reward, accuracy) or *diversity* (coverage, exploration), such as scientific discovery (Du et al., 2022).

**Sampling entropy.** For a given policy $p_\theta$, we define average token-level sampling entropy as

$$\mathcal{H}(\theta) = -\mathbb{E}_{\boldsymbol{x} \sim p_\theta} \left[ \frac{1}{|\boldsymbol{x}|} \sum_{t=1}^{|\boldsymbol{x}|} \log p_\theta(x_t \mid \boldsymbol{x}_{<t}) \right]. \tag{4}$$

This is the policy's intrinsic stochasticity per generated token and coincides with policy entropy in RL terminology (Cui et al., 2025).[1]

#### 2.1.2 MOLECULAR GENERATION

We ground the analysis primarily in molecular generation, a domain where both sample quality and diversity are essential and quantitatively measurable. Quality is given by a property scoring function (e.g., bioactivity estimators), while diversity is well-defined in chemical space. A common diversity metric is the mean pairwise Tanimoto distance (Tanimoto, 1958) over extended-connectivity fingerprints (ECFPs) (Rogers & Hahn, 2010) of molecules (Benhenda, 2018):

$$\mathrm{Div}(\{m_1, \dots, m_n\}) = \frac{1}{n(n-1)} \sum_{1 \leq i \neq j \leq n} d_T(m_i, m_j) \in [0, 1], \tag{5}$$

where $d_T$ is the Tanimoto distance. Molecular structures can be stringified as SMILES strings (Weininger, 1988), and SMILES-based language models finetuned with RL on property rewards achieve state-of-the-art results across standard benchmarks (Gao et al., 2022; He et al., 2024).

---

[1]*Clarification.* We distinguish (i) the *model (policy) entropy* $\mathcal{H}(\theta)$ in Equation 4, which averages over the model's own distribution, from (ii) *sample entropy* computed on a finite set of generated sequences. The former measures the model's uncertainty; the latter is a descriptive statistic of outputs.

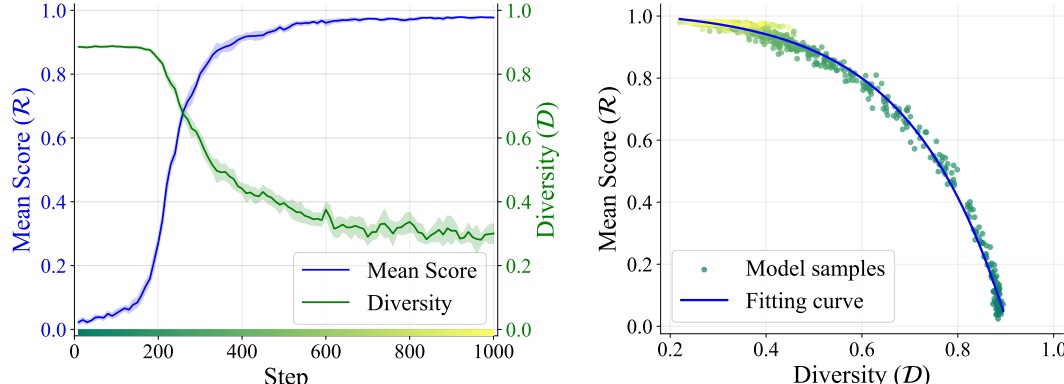

Figure 2: *Left:* Evolution of mean score ($\mathcal{R}$) and diversity ($\mathcal{D}$) across RL steps. *Right:* Relationship between $\mathcal{R}$ and $\mathcal{D}$, where each point corresponds to a set of molecules sampled from a model checkpoint. Colors denote RL steps as in the left plot. The curve is fitted by $\mathcal{R} = -a \cdot \exp(c \cdot \mathcal{D}) + b$.

## 2.2 FITTING THE CURVE BETWEEN QUALITY AND DIVERSITY

### 2.2.1 EXPERIMENTAL SETUP.

We train language models for molecular generation to investigate the relationship between quality and diversity of generated samples. The baseline model adopts the GPT architecture (Radford et al., 2019), consisting of 12 transformer layers with 12 attention heads and an embedding dimension of 384, totaling approximately 21M parameters. The model is pretrained on canonical SMILES strings from the ChEMBL dataset (Mendez et al., 2019) using an atom-wise tokenizer.

As the RL objective, we employ the JNK3 score, which estimates the bioactivity of a molecule against c-Jun N-terminal kinase 3 (JNK3), a target implicated in the treatment of Alzheimer's disease (Li et al., 2018). The scoring function is a random forest model trained on the ExCAPE-DB dataset (Sun et al., 2017), producing values in $[0, 1]$, where higher scores correspond to stronger predicted activity (Huang et al., 2021).

We finetune the pretrained model with reinforcement learning using the popular Reinvent algorithm (Olivecrona et al., 2017), running for 1000 RL steps. Importantly, we do not accumulate molecules sampled across RL steps, since our focus is on the model states during the RL process. To characterize these states, 100 checkpoints are uniformly selected throughout training. For each checkpoint, we perform 10 independent sampling runs of 100 molecules each. The resulting samples are used to compute the mean JNK3 score (quality) and molecular diversity, as well as to estimate the sampling entropy.

All experiments are conducted on NVIDIA A100 80GB GPUs. Further details of the experimental configuration are provided in Appendix A.

### 2.2.2 THE RELATIONSHIP BETWEEN QUALITY AND DIVERSITY

During iterative RL finetuning of language models, a common pattern emerges: output quality gradually improves while diversity decreases. This phenomenon is often described qualitatively as a "trade-off," in which the model transitions from low-quality/high-diversity states to high-quality/low-diversity ones. Figure 2 *Left* illustrates this process in our experiments, where generation quality rises from near 0 and converges toward 1, while molecular diversity declines from 0.88 to approximately 0.3.

Crucially, the quantitative form of this relationship has remained largely unexplored. As shown in Figure 2 *Right*, plotting mean score ($\mathcal{R}$) against diversity ($\mathcal{D}$) reveals a strikingly consistent pattern: points trace a curve from the lower-right (low $\mathcal{R}$, high $\mathcal{D}$) to the upper-left (high $\mathcal{R}$, low $\mathcal{D}$). The data fit the following exponential form with remarkable accuracy:

$$\mathcal{R} = -a \cdot \exp(c \cdot \mathcal{D}) + b, \tag{6}$$

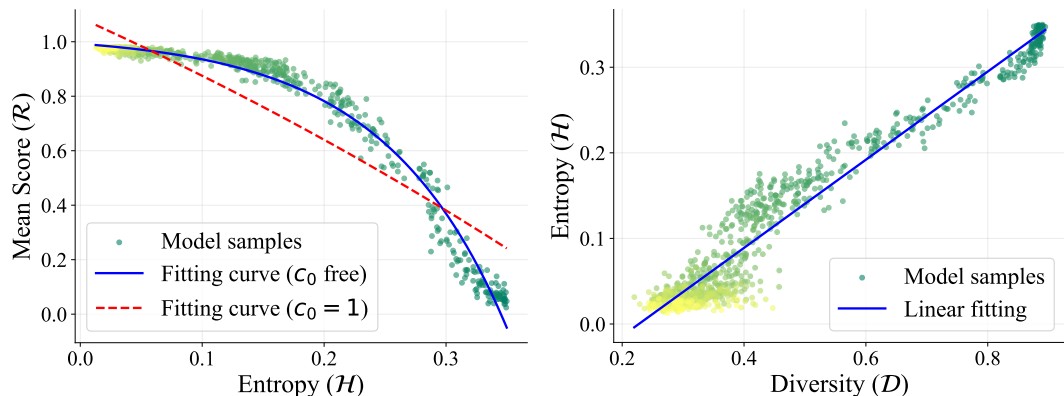

Figure 3: *Left:* Relationship between mean score ($\mathcal{R}$) and sampling entropy ($\mathcal{H}$). The blue curve corresponds to $\mathcal{R} = -a_0 \cdot \exp(c_0 \cdot \mathcal{H}) + b_0$, while the red dotted curve corresponds to $\mathcal{R} = -a_0 \cdot \exp(\mathcal{H}) + b_0$. *Right:* Relationship between sampling entropy ($\mathcal{H}$) and diversity ($\mathcal{D}$), with a linear fit. In both subfigures, each point represents a set of molecules sampled from a model checkpoint, and colors denote RL steps as in Figure 2 *Left*.

where $a, b, c > 0$ are fitted coefficients. Empirically, here we obtain $a = 0.0109 \pm 0.0008$, $b = 1.023 \pm 0.004$, and $c = 5.024 \pm 0.085$, achieving $R^2 = 0.996$.

Notably, this law is independent of step index in the RL process or training efficiency. Instead, it characterizes the expected trajectory of a model during finetuning for a fixed task: the initial point reflects the pretrained model's baseline ($\mathcal{R}_{\mathrm{pre}}, \mathcal{D}_{\mathrm{pre}}$), while the terminal point reflects the task-specific optimum ($\mathcal{R} = \mathcal{R}_{\mathrm{max}}, \mathcal{D} = 0$), which may not be fully reached in practice.

### 2.2.3 THE UNDERLYING ENTROPY MECHANISM

Although the empirical law in Equation 6 fits well with experimental results with only three parameters, its underlying mechanism warrants explanation. Entropy, a central concept in information theory, is known to correlate with diversity (Leinster, 2020). Prior work (Cui et al., 2025) reported an exponential law between quality ($\mathcal{R}$) and sampling entropy ($\mathcal{H}$):

$$\mathcal{R} = -a_0 \cdot \exp(\mathcal{H}) + b_0, \tag{7}$$

with $a_0, b_0 > 0$. This observation motivates us to probe the role of entropy in shaping the quality-diversity relationship.

Figure 3 presents two complementary analyses. First, the relation between $\mathcal{R}$ and $\mathcal{H}$ is better captured by introducing a scaling coefficient $c_0 > 0$:

$$\mathcal{R} = -a_0 \cdot \exp(c_0 \cdot \mathcal{H}) + b_0, \tag{8}$$

which significantly improves fit over Equation 7. Second, $\mathcal{H}$ is approximately linearly correlated with molecular diversity $\mathcal{D}$, suggesting an empirical bridge between entropy and diversity in molecular generation tasks.

Together, these results explain Equation 6:

$$\mathcal{R} = -a_0 \cdot \exp(c_0 \cdot \mathcal{H}) + b_0, \quad \mathcal{H} = a_1 \cdot \mathcal{D} + b_1,$$
$$\Rightarrow \quad \mathcal{R} = -a \cdot \exp(c \cdot \mathcal{D}) + b, \tag{9}$$

with $a = a_0 \exp(c_0 b_1)$, $b = b_0$, and $c = c_0 a_1$. Numerically, we find $a_0 = 0.0433 \pm 0.0041$, $b_0 = 1.040 \pm 0.007$, $c_0 = 9.086 \pm 0.267$, $a_1 = 0.518 \pm 0.008$, and $b_1 = -0.118 \pm 0.004$, fully consistent with the parameters in Equation 6. This establishes entropy as the hidden mechanism underpinning the observed exponential law.

### 2.3 UNDERSTANDING THE TRADE-OFF BETWEEN QUALITY AND ENTROPY

The fitted law in Equation 8 provides an empirical foundation for understanding the quality–diversity trade-off. A natural question is: why does an exponential form arise? Our goal in this subsection is

not to provide a rigorous derivation—which is intractable for deep neural generative models—but rather to offer a lightweight information-theoretic illustration that gives intuition for the structure of the law. Accordingly, we examine three conceptual questions:

1. Why does the relationship between quality ($\mathcal{R}$) and entropy ($\mathcal{H}$) take an exponential form?
2. Why is the scaling coefficient $c_0$ in Equation 8 necessary compared to Equation 7?
3. What determines the values of these coefficients?

To build intuition, consider sampling a token from the vocabulary for simplicity, which is the elementary unit of autoregressive generation. Let $T$ denote the typical set of the model distribution. Standard information-theoretic arguments suggest that its size grows as $|T| = \exp(\mathcal{H})$ (MacKay, 2003). Let $S$ be the "good" subset of desirable tokens, and for illustration, define $U$ and $Q$ as the uniform distributions on $T$ and $S \cap T$, respectively. The expected quality can be expressed as the maximum possible quality $\mathcal{R}_{\max}$ (when $T \subseteq S$) minus a divergence term:

$$\mathcal{R} = \mathcal{R}_{\max} - \lambda \, \mathbf{D}_{\chi^2}(Q \parallel U) = \mathcal{R}_{\max} - \lambda \sum_{x \in T} U(x) \left( \frac{Q(x)}{U(x)} - 1 \right)^2$$

$$= \mathcal{R}_{\max} + \lambda - \lambda \frac{|T|}{|S \cap T|},$$

(10)

where $\mathbf{D}_{\chi^2}$ is the chi-square divergence (Nishiyama & Sason, 2020) (derivation in Appendix B). This expression is not intended as an exact model of deep generative behavior, but it illustrates a simple mechanism: training simultaneously shrinks the typical set (reducing entropy) and increases its overlap with high-quality regions.

During RL finetuning, $|T|$ typically decreases as the model becomes more deterministic, while alignment toward desirable outputs can be captured by a coarse approximation $|S \cap T| \asymp |T|^\eta$ for some $\eta \le 1$. This yields

$$\mathcal{R} = \mathcal{R}_{\max} + \lambda - \lambda |T|^{1-\eta} = -\lambda \cdot \exp\big((1-\eta)\mathcal{H}\big) + \mathcal{R}_{\max} + \lambda,$$

(11)

which matches the structure of Equation 8, with $a_0 = \lambda$, $b_0 = \mathcal{R}_{\max} + \lambda$, and $c_0 = 1 - \eta$.

This formulation highlights why $c_0$ is essential. When $c_0 = 1$ ($\eta = 0$), $|S \cap T|$ is nearly fixed, corresponding to non-exploratory finetuning as in Cui et al. (2025), where the good subset is already covered by the pretrained model. In contrast, exploration-heavy tasks such as molecular generation yield $\eta < 0$ and thus $c_0 > 1$, since $|S \cap T|$ expands as $|T|$ contracts.

Moreover, Equation 11 explains the fitted coefficients. At extreme convergence ($\mathcal{H} = 0$, $|T| = 1$), the model deterministically selects a single token in $S$, achieving $\mathcal{R}_{\max} = b_0 - a_0$. This is consistent with our empirical fits in Figure 3 *Left*, where $\mathcal{R}_{\max} = 1$. Moreover, $c_0 = 1 - \eta$ quantifies the balance between model convergence and exploratory alignment.

**Extension to sequence-level generation.** While the above analysis is framed at the token level for simplicity, the same reasoning extends to sequence-level generation. In this case, the typical set $T$ corresponds to the set of likely sequences under the model distribution. The "good" subset $S$ then represents sequences achieving high quality (scores). The same divergence-based argument applies: as training progresses, $T$ contracts while $S \cap T$ expands, producing an exponential relationship between expected score and entropy at the sequence level. This suggests that the entropy-based mechanism is not confined to molecular generation, but may also underlie quality-diversity trade-offs observed in broader language modeling tasks, such as the circle and line construction tasks we later study in Section 3.2.

## 3 UTILIZING THE QUALITY-DIVERSITY TRADE-OFF

After establishing a quantitative relationship between quality ($\mathcal{R}$), diversity ($\mathcal{D}$), and sampling entropy ($\mathcal{H}$), and revealing an entropy-based mechanism, we now turn to *applications*: how can one increase output diversity as much as possible under a target quality level during RL finetuning of language models? This section analyzes which factors shape the trade-off, how they act in practice, and how the fitted law can be applied.

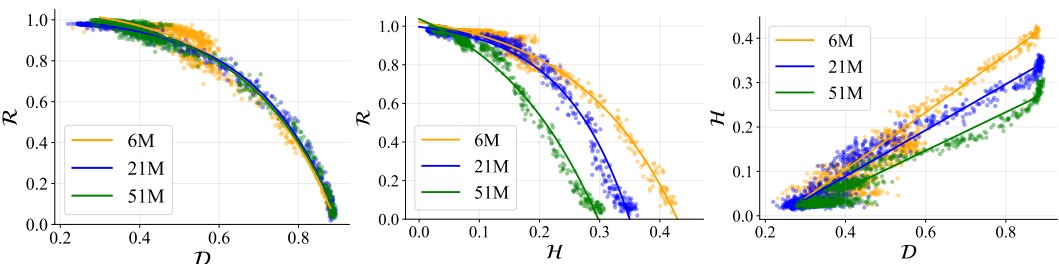

Figure 4: Influence of model scaling on relationships among mean score ($\mathcal{R}$), diversity ($\mathcal{D}$), and sampling entropy ($\mathcal{H}$).

### 3.1 INFLUENCING FACTORS

We conduct ablations under the experimental setup of Section 2.2.1, holding the evaluation protocol fixed while varying one factor at a time. We focus on three classes of factors: model scaling, reward transformations, and training setup (pretraining and RL finetuning).

#### 3.1.1 MODEL SCALING

Model size is a primary determinant of representation capacity and pretraining fit. Beyond the 21M-parameter model in Section 2.2, we repeat the analysis with 6M and 51M parameter variants that share the same model architecture, RL objective, and pipeline. The outcomes are demonstrated in Figure 4. All three models reach $\mathcal{R}_{\max} \approx 1$ under sufficient finetuning, demonstrating that, for this task, capacity is not the limiting factor once above a modest threshold.

At initialization of post-pretraining, the models exhibit similar diversity levels but different entropies: larger models have lower $\mathcal{H}$, consistent with their improved fit to the pretraining distribution. Consequently, their trajectories through $\mathcal{R}$–$\mathcal{H}$ space diverge, particularly in the early and middle stages of RL finetuning. However, the $\mathcal{R}$–$\mathcal{D}$ trajectories remain tightly aligned across sizes: for a fixed task/reward and diversity metric, the attainable frontier is nearly invariant to model size once all models can approach $\mathcal{R}_{\max}$. In other words, scaling predominantly changes how quickly and through which entropy levels the model traverses the $\mathcal{R}$–$\mathcal{D}$ trajectory, rather than the trajectory itself.

#### 3.1.2 REWARD SHAPING

Reward shaping is commonly used to stabilize or accelerate RL finetuning for language models. Building on the JNK3 score from Section 2.2, we examine two representative transformations: a linear map and a monotonic nonlinear map,

$$f_{\text{linear}}(x) = \frac{x}{2}, \qquad f_{\text{nonlinear}}(x) = x^2, \tag{12}$$

and use the transformed values as the RL reward while monitoring quality via the original score.

Figure 5 shows that a linear scaling has a negligible effect on either $\mathcal{R}$–$\mathcal{D}$ or $\mathcal{R}$–$\mathcal{H}$ when quality is plotted in the original units: it can be absorbed into $(\mathcal{R}_{\max}, \lambda)$ in Equation 10 without altering the divergence term that governs the shape of the curve. In contrast, the nonlinear map noticeably changes the trajectory in $\mathcal{R}$–$\mathcal{H}$ by reweighting the marginal gain of entropy reduction at different reward levels. Yet, the $\mathcal{R}$–$\mathcal{D}$ relation remains remarkably stable across the three settings, suggesting that the trajectory is dictated primarily by the task distribution and the chosen diversity metric, rather than the exact monotonic transformation of the reward.

#### 3.1.3 TRAINING SETUP

We separately discuss the effects of pretraining (which fixes the initial point) and RL finetuning (which determines the training dynamics).

**Pretraining.** As illustrated in Figure 1, pretraining sets $(\mathcal{R}_{\text{pre}}, \mathcal{D}_{\text{pre}}, \mathcal{H}_{\text{pre}})$. Data choice, augmentation, and entropy-oriented regularization can shift this initial point rightward by increasing diversity

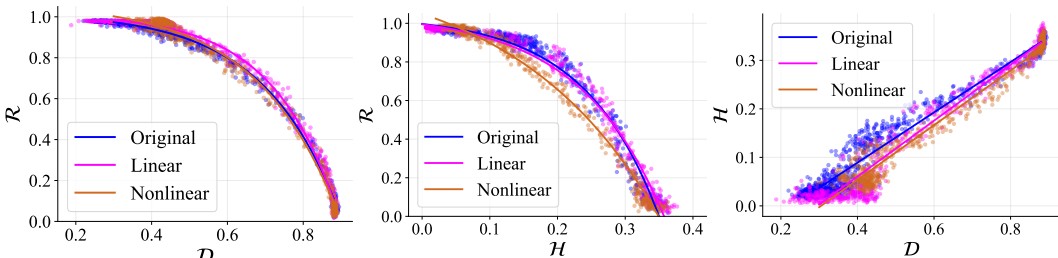

Figure 5: Influence of score transformations on relationships among mean score ($\mathcal{R}$, measured by the original JNK3 score), diversity ($\mathcal{D}$), and sampling entropy ($\mathcal{H}$).

at comparable baseline quality, thereby improving the downstream trade-off before any RL updates. Where possible, higher-quality and more diverse pretraining corpora move the model closer to the desirable region in $\mathcal{R}$–$\mathcal{D}$ at the outset.

**RL finetuning.** Different RL algorithms and hyperparameters chiefly influence efficiency and, in some cases, the ceiling attainable $\mathcal{R}_{\text{max}}$. Empirically and conceptually, once the task (reward) and initial/terminal points are fixed, the $\mathcal{R}$–$\mathcal{D}$ trajectory appears algorithm-agnostic, which was also observed by Cui et al. (2025) on $\mathcal{R}$–$\mathcal{H}$; what changes is the parameterization of points along it.[2]

In summary, for a fixed model family and task, the factors under control are (i) the *initial* point $(\mathcal{R}_{\text{pre}}, \mathcal{D}_{\text{pre}})$ via pretraining and (ii) the *ceiling* $\mathcal{R}_{\text{max}}$ via finetuning. The shape of the $\mathcal{R}$–$\mathcal{D}$ curve itself reflects constraints imposed by the generative task and diversity definition.

### 3.2 DIFFERENT TASKS AND DOMAINS

Thus far, all experiments have used the JNK3 objective. We now evaluate whether the same quality–diversity and quality–entropy relationships persist across substantially different tasks, spanning both molecular and non-molecular domains.

#### 3.2.1 DIFFERENT MOLECULAR OBJECTIVES

We first replicate the protocol on two additional molecular property objectives: (1) GSK3b score, which estimates the bioactivity of a molecule against the Glycogen synthase kinase 3 beta target (GSK3b) (Li et al., 2018), and (2) QED score, which quantitatively estimates the drug-likeness of a molecule (Bickerton et al., 2012). Both scores range in $[0, 1]$, and higher scores indicate a more desirable property. The pretrained 21M model is reused and the RL finetuning setup is kept unchanged to isolate task effects.

Figure 6 shows that the exponential relationships persist for both $\mathcal{R}$–$\mathcal{D}$ and $\mathcal{R}$–$\mathcal{H}$ across the new tasks, albeit with task-specific ranges and fitted coefficients. This supports *generality* within chemical space: the quantitative law is not peculiar to the JNK3 score but emerges across distinct objectives. At the same time, the variation in coefficients highlights *task uniqueness*: the attainable diversity at a fixed quality depends on the reward landscape and the desirable sample distribution relevant to the objective.

#### 3.2.2 TEXTUAL EXPLORATORY CREATIVITY TASKS

To test whether our findings extend beyond molecular generation, we further evaluate two structured creativity tasks introduced by Nagarajan et al. (2025): *circle construction* and *line construction*. In these tasks, a sequence of characters encodes a set of directed edges, where each adjacent pair $(x, y)$ represents an edge $x \rightarrow y$. A string forms a *circle* if these edges create a single simple cycle without repetition, and it forms a *line* if they create a simple directed path without cycles. These tasks provide explicit, domain-appropriate notions of *quality* (validity of the constructed structure) and

---

[2]We treat temperature/top-$k$/top-$p$ as decoding controls, not finetuning hyper-parameters, because they alter $(\mathcal{R}_{\text{pre}}, \mathcal{H}_{\text{pre}}, \mathcal{D}_{\text{pre}})$ for a fixed policy at sample time.

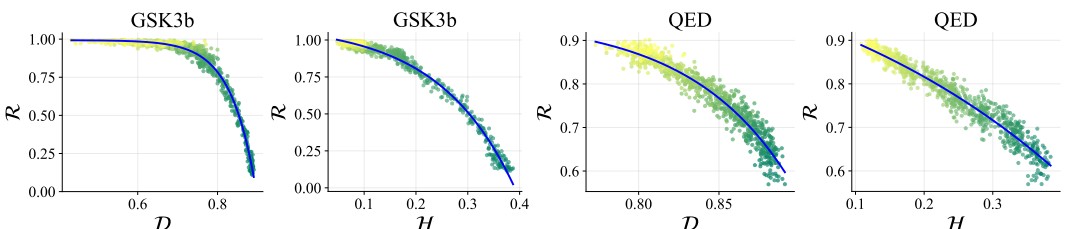

Figure 6: Relationships of $\mathcal{R}$ vs. $\mathcal{D}$ and $\mathcal{R}$ vs. $\mathcal{H}$ on GSK3b and QED objectives. The configurations follow Figures 2 *Right* and 3 *Left*.

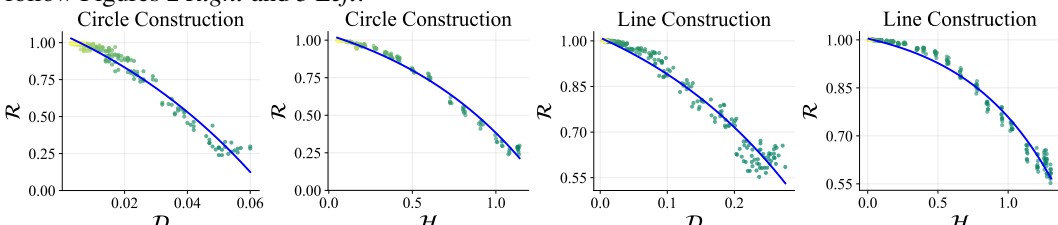

Figure 7: Relationships of $\mathcal{R}$ vs. $\mathcal{D}$ and $\mathcal{R}$ vs. $\mathcal{H}$ on circle construction and line construction.

*diversity* (structural variation in the induced graphs), enabling quantitative analysis analogous to the molecular setting.

We finetune GPT2-XL (1.5B parameters) (Radford et al., 2019) with the proximal policy optimization (PPO) algorithm (Schulman et al., 2017) on both tasks and compute quality, diversity, and sampling entropy at uniformly spaced RL checkpoints. As shown in Figure 7, the same exponential forms reappear, achieving $R^2 > 0.97$ on both tasks. The trajectories exhibit the same qualitative shape as in molecular generation, despite differences in model size, domain, and reward semantics.

### 3.3 DISCUSSION

We synthesize practical guidance for RL finetuning language models in exploratory scenarios where both high quality and high diversity are desired:

1. **Prioritize the initial point and the ceiling quality.** The position of the pretrained model $(\mathcal{R}_{\mathrm{pre}}, \mathcal{D}_{\mathrm{pre}})$ strongly influences the downstream frontier. Improvements in pretraining— including higher-quality data, broader coverage, and better model capacity—shift this initial point toward more favorable regions. Likewise, finetuning techniques that raise the achievable ceiling $\mathcal{R}_{\max}$ expand the reachable end of the trajectory.

2. **Do not expect reward shaping or algorithm swaps to move the trajectory.** Ablations show that monotonic transformations of the reward, as well as different RL algorithms, primarily affect how the model moves along the trajectory (i.e., training efficiency), rather than the trajectory itself. The frontier is largely governed by the task distribution and the chosen diversity metric, consistent with the entropy-based interpretation in Section 2.3.

3. **Use mixtures to exceed single-policy diversity.** When a single policy saturates the quality-diversity frontier, mixtures of policies or multi-agent ensembles can achieve higher realized diversity at a fixed target quality. Such methods may help cover a broader portion of the desirable distribution than any single RL-trained policy (Hu et al., 2024).

Beyond decision support, the fitted exponential law also serves as a *predictor*: given a target $\mathcal{R}$, it estimates the achievable $\mathcal{D}$ (and vice versa), particularly aiding in creative scenarios. In practice, we recommend reporting the fitted coefficients and their confidence intervals for robust estimation.

## 4 RELATED WORKS

Reinforcement learning (RL) finetuning has become the dominant paradigm for post-training autoregressive language models. However, optimizing for quality often sharpens the model's output

distribution and reduces diversity. Li et al. (2025a) documents that RL post-training prioritizes quality at the expense of diversity, limiting the range of ideas and degrading performance in creative and exploratory tasks. Similarly, Anschel et al. (2025) notes that RL improves generation factuality and alignment but often reduces output diversity, leading to mode collapse. Although the existence of a quality-diversity trade-off during RL finetuning of language models is widely acknowledged, its *quantitative* form has been obscure.

Quality-diversity (QD) algorithms such as novelty search (Lehman & Stanley, 2011) and MAP-Elites (Mouret & Clune, 2015) have been introduced in evolutionary computation to generate large collections of diverse, high-performing solutions and have been applied in robotics, engineering, and game design (Pugh et al., 2016). Yet, QD methods require hand-crafted behavioral descriptors, making them difficult to transfer to language models.

Prior works attempt to mitigate diversity collapse of language models either by modifying the training objective or the decoding strategy, including penalizing repetition to increase lexical variety (Welleck et al., 2019), matching to high-entropy distributions (Zhang et al., 2024), and applying sparse logit updates (Li et al., 2024). Particularly for reinforcement learning, diverse preference optimization (DivPO) selects rare but high-quality responses from a pool and reports substantial diversity gains while maintaining quality (Lanchantin et al., 2025), and group-aware policy optimization (GAPO) extends Group Relative Policy Optimization by rewarding uniform sampling over valid completions, thereby reducing mode collapse without degrading accuracy (Anschel et al., 2025). However, none of these approaches explores the quantitative relationship between quality and diversity, limited to giving qualitative descriptions (Kirk et al., 2023; Murthy et al., 2025). Notably, Cui et al. (2025) observes an exponential relationship between RL reward and sampling entropy of reasoning language models, which addresses a quality-entropy trade-off but does not consider actual sample diversity.

## 5 CONCLUSION

In this paper, we move beyond qualitative descriptions of diversity collapse in reinforcement learning of language models and provide a quantitative characterization of the quality-diversity trade-off. Using molecular generation as a primary domain—where diversity is intrinsic and directly measurable—we show that RL checkpoints follow a robust exponential law between quality and diversity, and establish sampling entropy as the hidden mechanism underpinning this trajectory. The companion exponential law between quality and entropy, together with the approximate linearity between entropy and diversity, explains the emergence of the observed frontier. Beyond molecular generation, we further validate these laws on two textual exploratory creativity tasks (circle construction and line construction) using GPT2-XL finetuned with PPO, demonstrating that the same relationships persist in a non-chemical, large-model setting. Finally, we show how the fitted laws can be translated into practical tools for predicting attainable diversity at target quality and informing exploratory RL workflows.

**Limitations.** Despite these advances, our study has limitations. First, although we include experiments on GPT2-XL for structured textual creativity tasks, we do not yet evaluate modern frontier-scale language models or open-ended natural language generation, where defining domain-appropriate quality and diversity metrics remains challenging (Tevet & Berant, 2020; Estève et al., 2025). Second, while the entropy-diversity relation appears approximately linear in both molecular and structured textual tasks, its robustness under alternative diversity metrics and across broader domains requires further investigation.

**Future directions.** Several directions are promising. One is to extend the analysis to emerging architectures such as diffusion-based language models and multi-modal models (Li et al., 2025b; Yin et al., 2024). Another is to conduct deeper evaluations of methods designed to promote diversity, such as policy ensembles or mixture-of-experts approaches. Finally, an open theoretical question is how the task-dependent quality–diversity Pareto frontier interacts with the model's actual RL trajectory; understanding this gap would deepen the theoretical foundations and further improve practical control of diversity in exploratory applications.

**Ethics Statement.** This work adheres to the ICLR Code of Ethics. Our study is theoretical and empirical in nature, focusing on the quality-diversity trade-off in reinforcement learning for language models. It does not involve human subjects, personal or sensitive data, or tasks with direct societal risks. The molecular generation tasks used in our experiments rely on standard open-source benchmarks and well-established evaluation metrics, without producing deployable compounds or actionable medical claims. We have taken care to avoid potential harms such as data leakage, privacy violations, or biased analyses. To the best of our knowledge, this research poses no ethical, legal, or security concerns beyond the usual scientific standards of rigor and reproducibility.

**Reproducibility Statement.** The complete codebase for experiments in this paper is provided in the supplementary material and will be released publicly upon publication. The main text describes the experimental pipeline, model configurations, and evaluation metrics, while Appendix A contains additional implementation details and hyperparameters. All datasets used are publicly available, and we clearly describe data preprocessing and scoring functions in the experimental setup.

**LLM Usage Statement.** Large language models (LLMs) are used solely as assistive tools for improving the clarity of writing and for generating auxiliary code snippets. They do not contribute to research ideation, design of experiments, or analysis of results. The human authors take full responsibility for all content, and no parts of the paper rely on unverifiable or fabricated outputs from LLMs.

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

## A    EXPERIMENT DETAILS

### A.1    MODEL DETAILS

We use a GPT-style, decoder-only Transformer with causal self-attention, token and positional embeddings, pre-norm residual blocks, GELU activations, and a two-layer feed-forward network (hidden size $\approx 4 \times d_{\text{model}}$). The output projection (LM head) is tied with the input token embeddings. Table 1 summarizes the three model variants used in our experiments.

Table 1: Transformer architecture variants.

| Variant | #Layers | #Heads | $d_{\text{model}}$ |
|---|---|---|---|
| Small (6.37M) | 8 | 8 | 256 |
| Base (21.38M) | 12 | 12 | 384 |
| Large (50.55M) | 16 | 16 | 512 |

### A.2    TRAINING DETAILS

**Pretraining (MLE).**    Models are pretrained with next-token prediction on a corpus of canonical SMILES. We use an atom-level vocabulary tailored to SMILES with special tokens `[BOS]`, `[EOS]`, and `[PAD]`. The maximum sequence length is 128 (tokens). Optimization uses AdamW with $(\beta_1, \beta_2) = (0.9, 0.95)$, weight decay 0.1, initial learning rate 0.001, cosine decay with 1% warm-up steps, global batch size 1024, and dropout $p = 0.1$. Training proceeds for 20 epochs on 1 NVIDIA A100 80GB GPU.

**RL finetuning.**    Starting from the pretrained checkpoint (the *prior*), we finetune the policy with policy-gradient updates to maximize the JNK3 oracle reward (or other molecular property objectives). At each iteration, we sample 128 rollouts of length up to 128 using temperature 1.0 and top-$k$=10, then compute rewards $r(x)$ from the scoring function. The RL loss function follows Reinvent Olivecrona et al. (2017). Sampling validity is enforced by RDKit validation, and deduplication uses canonical SMILES.

For more experimental setup details, please refer to the code in the supplementary material.

### A.3    OUTPUT EXAMPLES

Three example molecules produced by the RL pipeline with JNK3 score $= 1$ (perfect according to our oracle) are shown in Figure 8. Despite high reward, all three share an *identical* substructure (on the right parts), leading to low structural diversity. This illustrates the quality-diversity trade-off discussed in the main text: reward optimization can collapse diversity.

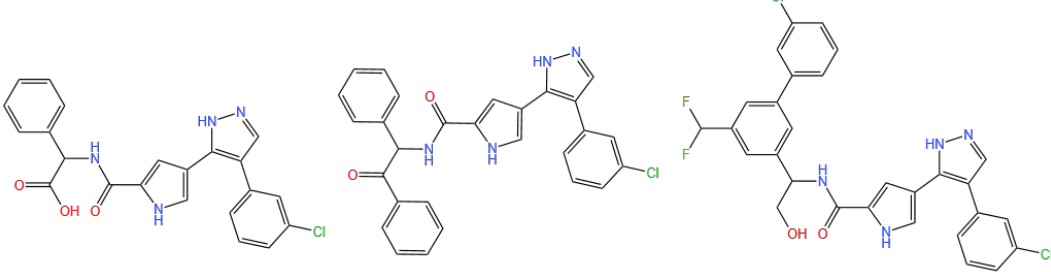

Figure 8: Three example molecules produced by the RL pipeline with JNK3 score $= 1$.

# B    Theoretical Derivations

## B.1    Derivation of Equation 9

Substituting the linear relationship $\mathcal{H} = a_1 \cdot \mathcal{D} + b_1$ into Equation 8 gives

$$
\begin{aligned}
\mathcal{R} &= -a_0 \cdot \exp(c_0(a_1\,\mathcal{D} + b_1)) + b_0 \\
&= -a_0 \cdot \exp(c_0 b_1) \cdot \exp(c_0 a_1 \cdot \mathcal{D}) + b_0 \\
&= -\underbrace{\left[a_0\,\exp(c_0 b_1)\right]}_{=a} \exp\cdot(\underbrace{c_0 a_1}_{=c}\cdot\mathcal{D}) + \underbrace{b_0}_{=b}.
\end{aligned}
$$

$$
\Rightarrow \quad a = a_0\,\exp(c_0 b_1), \qquad c = c_0 a_1, \qquad b = b_0,
$$

in the asserted form of Equation 6:

$$
\mathcal{R} = -a \cdot \exp(c \cdot \mathcal{D}) + b.
$$

## B.2    Derivation of Equation 10

$$
\begin{aligned}
\mathcal{R} &= \mathcal{R}_{\max} - \lambda\,\mathbf{D}_{\chi^2}(Q \parallel U) = \mathcal{R}_{\max} - \lambda \sum_{x \in T} U(x)\left(\frac{Q(x)}{U(x)} - 1\right)^2 \\
&= \mathcal{R}_{\max} - \lambda \sum_{x \in T}\left(\frac{Q(x)^2}{U(x)} - 2Q(x) + U(x)\right) \\
&= \mathcal{R}_{\max} - \lambda\left[\sum_{x \in T}\frac{Q(x)^2}{U(x)} - 2\sum_{x \in T}Q(x) + \sum_{x \in T}U(x)\right] \\
&= \mathcal{R}_{\max} - \lambda\left[\sum_{x \in S \cap T}\frac{(1/|S \cap T|)^2}{1/|T|} - 2 \cdot 1 + 1\right] \\
&= \mathcal{R}_{\max} - \lambda\left[\frac{|T|}{|S \cap T|} - 1\right] = \mathcal{R}_{\max} + \lambda - \lambda\frac{|T|}{|S \cap T|}.
\end{aligned}
$$

## B.3    Information-Theoretic Notation Used in Section 2.3

This subsection briefly introduces information theoretic notions used in Section 2.3: the *typical set* and the symbol $\asymp$, both of which appear in our information-theoretic illustration.

**Typical set.**    For a discrete distribution $p(x)$ with Shannon entropy

$$
\mathcal{H}(p) = -\sum_x p(x)\log p(x),
$$

the *typical set* $T$ is the collection of outcomes whose log-probabilities concentrate around the entropy:

$$
T = \left\{x : -\frac{1}{n}\log p(x) \approx \mathcal{H}\right\}.
$$

A classical result in information theory states that, under mild conditions, most of the total probability mass lies in $T$, and its size grows approximately as

$$
|T| \approx \exp(\mathcal{H}).
$$

This intuitive connection—higher entropy corresponds to a larger effective support—is the only property of the typical set invoked in Section 2.3.

**The symbol $\asymp$.** The notation $f \asymp g$ denotes that two quantities are *proportional up to a constant factor*, i.e.,

$$f \asymp g \iff \exists\, c_1, c_2 > 0 \text{ such that } c_1 g \leq f \leq c_2 g.$$

It is weaker than equality and does not specify the constants. In Section 2.3, we use

$$|S \cap T| \asymp |T|^{\eta}$$

to express a coarse, empirical alignment trend: as the model becomes more deterministic (entropy decreases and $|T|$ shrinks), the portion of the typical set that corresponds to desirable outcomes grows roughly like a power of $|T|$. This relation is used only for intuition and is not assumed to be an exact model of the trained distribution.

