# OpenReview forum: "Exploring the Trade-off between Quality and Diversity of Language Models during Reinforcement Learning"
_ICLR.cc/2026/Conference — Submitted to ICLR 2026_

### Official Review · Reviewer_ZnyY · 2025-10-25

**Soundness:** 2
**Presentation:** 2
**Contribution:** 2
**Rating:** 2
**Confidence:** 4

**Summary:**

This paper explores the relationship between quality and diversity in molecular generation, proposing an exponential law between the two.

**Strengths:**

The paper employs a meaningful, task-dependent diversity metric to construct and validate the proposed relationship.

Comprehensive experiments are conducted across models of different sizes and under varying reward-shaping mechanisms.

The analysis spans multiple datasets within the same domain, enhancing the robustness of the empirical findings.

**Weaknesses:**

My primary concern is whether the paper's conclusions are, as claimed, general for language models. While I appreciate the authors’ use of domain-related metrics, the meaning of diversity can vary drastically across tasks. In the theoretical analysis, the paper assumes—without sufficient justification—that molecular diversity and sampling entropy are linearly related. This assumption significantly limits the generalizability of the proposed conclusions to other tasks or diversity metrics.

And some additional unexamined assumptions include:

The assumption of uniform distributions over subsets $U$ and $Q$ is unrealistic for trained language models.

The approximation $|S\cap T|\approx |T|^\eta$
, which lacks theoretical grounding, and the connection between $\eta$ and actual training dynamics remains unclear.

Consequently, the “information-theoretic explanation,” while intuitively appealing, is not mathematically rigorous—it serves more as a heuristic argument than a formal derivation.

From the RL algorithm chosen, the authors employ standard RL methods in molecular generation, which differ from reinforcement learning methods commonly used in current language models (e.g., GRPO, PPO). The treatment of regularization and entropy loss plays a crucial role here. Validating the proposed law across a broader set of reinforcement learning algorithms would strengthen the claim.

Overall, the paper frequently uses “language model” in a general sense, which may overextend the applicability of the proposed exponential law.

**Questions:**

Can authors give more explanation on why $\eta$ remains constant throughout the entire training process?

---

> ### Author Response · Authors · 2025-11-21
> **Rebuttal (1/2)**
>
> Thank you for the thoughtful and detailed review. We address each concern point-by-point below.
>
> > My primary concern is whether the paper's conclusions are, as claimed, general for language models. While I appreciate the authors’ use of domain-related metrics, the meaning of diversity can vary drastically across tasks.
>
> We appreciate the concern regarding generality. To directly test whether the exponential laws hold beyond SMILES-based molecular generation where diversity definitions are domain-specific, we added new experiments on exploratory tasks with large natural language models. Specifically, we adopt the **circle construction** and **line construction** tasks from Nagarajan et al. (ICML 2025) [1], where both quality and diversity/creativity are clearly defined.
>
> We finetuned **GPT2-XL (1.5B parameters)** using **PPO**. As shown in **Figure 7 (page 8)** of the revised manuscript, both $\mathcal{R}-\mathcal{D}$ and $\mathcal{R}-\mathcal{H}$ again exhibit excellent exponential fits ($R^2 > 0.97$). These results demonstrate that the observed phenomenon is **not** restricted to molecular space or small models, and provide concrete evidence that the exponential frontier extends to text LMs and non-chemical diversity metrics.
>
> [1] Nagarajan et al., [Roll the dice & look before you leap: Going beyond the creative limits of next-token prediction](https://openreview.net/forum?id=Hi0SyHMmkd), ICML 2025 (oral).
>
> > In the theoretical analysis, the paper assumes—without sufficient justification—that molecular diversity and sampling entropy are linearly related. This assumption significantly limits the generalizability of the proposed conclusions to other tasks or diversity metrics.
>
> You are correct that entropy–diversity linearity need not hold universally, especially across very heterogeneous tasks. We clarify that:
>
> - We **do not assume** this linearity as a theoretical axiom.
> - Instead, we **empirically observe** it in molecular generation (see Figure 3 *Right*), and use it as an empirical bridge to explain why the entropy-based law composes into the diversity-based law, as stated in Section 2.2.
>
> For other domains, the $\mathcal{R}-\mathcal{H}$ exponential law remains the more fundamental relationship. Different diversity metrics may yield different degrees of linearity with entropy, but this does not affect the $\mathcal{R}-\mathcal{H}$ law nor the main conclusions of the paper. We have strengthened this clarification in the revised text.

---

> > ### Author Response · Authors · 2025-11-21
> > **Rebuttal (2/2)**
> >
> > > And some additional unexamined assumptions include:
> > >
> > > 1. The assumption of uniform distributions over subsets $U$ and $Q$ is unrealistic for trained language models.
> > >
> > > 2. The approximation $|S \cap T| \approx |T|^{\eta}$, which lacks theoretical grounding, and the connection between $\eta$ and actual training dynamics remains unclear.
> > >
> > > Consequently, the “information-theoretic explanation,” while intuitively appealing, is not mathematically rigorous—it serves more as a heuristic argument than a formal derivation.
> > >
> > > Question: Can authors give more explanation on why $\eta$ remains constant throughout the entire training process?
> >
> > We agree that several assumptions in Section 2.3 are idealized. As stated in the opening paragraph of that section, our goal is to provide **theoretical intuition** rather than a mathematically rigorous derivation, which is currently intractable for deep models.
> >
> > Nevertheless, we would like to justify our modeling choices:
> >
> > 1. The use of uniform distributions over the typical set is a **standard information-theoretic simplification**, not a literal claim about trained LLMs, and even if this assumption is relaxed, the chi-square divergence structure in Equation (10) continues to induce an **exponential dependence on entropy**, leaving the overall form of the law robust.
> >
> > 2. The approximation $|S \cap T| \asymp |T|^{\eta}$ is introduced **as an analytic device** to express the relative rate at which the “good’’ subset aligns with the typical set during RL finetuning. We explicitly acknowledge in the revision that a full theoretical grounding for this approximation is currently unavailable and constitutes an interesting direction for future work.
> > 3. Regarding your question on the constancy of $\eta$, we note that this parameter is **not assumed** to be constant; rather, its empirical manifestation through the fitted coefficient $c_0 = 1 - \eta $ remains **approximately stable across RL checkpoints** in all of our experiments—including both molecular tasks and the newly added GPT2-XL experiments. We clarified this empirical nature of $\eta$ in the revised manuscript and emphasized that it should be interpreted as a useful descriptive parameter rather than a theoretically immutable constant.
> >
> > > From the RL algorithm chosen, the authors employ standard RL methods in molecular generation, which differ from reinforcement learning methods commonly used in current language models (e.g., GRPO, PPO). The treatment of regularization and entropy loss plays a crucial role here. Validating the proposed law across a broader set of reinforcement learning algorithms would strengthen the claim.
> >
> > We agree that validating across RL algorithms strengthens generality. In response to the your comment, our **new experiments use PPO**, the dominant algorithm in LLM post-training. The updated **Figure 7** shows that the exponential laws persist under PPO on non-molecular LLM tasks.
> >
> > Regarding entropy regularization: Cui et al. (2025) [2] report that explicit entropy bonuses are **highly unstable and degrade RL performance on language models**, making them unsuitable as ablations in our setting.
> >
> > [2] Cui et al., [The Entropy Mechanism of Reinforcement Learning for Reasoning Language Models](https://arxiv.org/abs/2505.22617), 2025.
> >
> > > Overall, the paper frequently uses “language model” in a general sense, which may overextend the applicability of the proposed exponential law.
> >
> > We have revised wording throughout to clearly distinguish between molecular language models (SMILES-based autoregressive models) and large natural language models. The paper now position the exponential laws as **empirically supported across multiple domains**, with careful discussion of scope and limitations.
> >
> > ------
> >
> > We thank you again for these insightful comments. We believe that the substantial new LLM experiments, the clarifications of the theoretical scope, and the expanded explanations of modeling assumptions directly address the concerns raised, and we hope they will positively inform your reassessment of our submission.

---

> > > ### Author Response · Authors · 2025-11-26
> > > **Follow-Up on Rebuttal Before Discussion Deadline**
> > >
> > > Dear Reviewer ZnyY,
> > >
> > > Thank you for your detailed review and valuable comments. As the discussion deadline is approaching, I would like to kindly invite you to take a look at our rebuttal and the additional experiments/clarifications we provided in response to your questions. We believe these directly address the concerns you raised.
> > >
> > > If any part remains unclear, we are happy to further clarify within the remaining time. We would greatly appreciate your updated assessment after considering the rebuttal.
> > >
> > > Thank you again for your time and constructive feedback.
> > >
> > > Authors

---

> > > > ### Comment · Reviewer_ZnyY · 2025-11-26
> > > >
> > > > I appreciate the authors' efforts in adding experiments, which have a high computation cost. However, while the inclusion of a task beyond molecular generation is appreciated, the evidence base of only two domains remains too narrow to robustly support the paper's claimed empirical relationship between performance and diversity for RL-trained language models. Simply adding one more task does not convincingly demonstrate that this relationship holds broadly. I encourage the authors to either frame this finding within a more specific and well-scoped context (e.g., focusing solely on molecule generation) – which would not diminish the contribution – or to provide clearer theoretical intuition. I would appreciate it if the theoretical intuition could help readers predict for which tasks or diversity metrics this relationship would manifest. Given the current scale of evidence compared with the current claim and theory, I believe the paper is better suited for submitting to a workshop, and I prefer to maintain my current score.

---

> > > > > ### Author Response · Authors · 2025-12-01
> > > > > **Further Rebuttal**
> > > > >
> > > > > Thank you again for the follow-up and for engaging deeply with our work. We respectfully clarify a few points regarding the scope of our experiments and the framing of our claims.
> > > > >
> > > > > **1. On the expectation of broader empirical coverage.**
> > > > > We fully agree that demonstrating generality across many domains would be ideal. However, as we emphasized throughout the revision, most natural-language tasks do not admit a meaningful or quantifiable notion of diversity that is compatible with our theoretical analysis (e.g., Tanimoto distances in molecules or the discrete structural creativity metrics in circle/line construction). This severely limits the space of tasks where the quality–diversity frontier can even be defined, let alone measured consistently.
> > > > >
> > > > > Within the set of *quantifiable* tasks available, we now evaluate:
> > > > >
> > > > > - **Molecular generation** across multiple datasets, model scales, and RL reward mechanisms
> > > > > - **Large language models** (GPT-2-XL) on **non-molecular** structured-generation tasks with clearly defined quality and diversity metrics
> > > > >
> > > > > These additions expand the empirical base from one domain to a structurally different autoregressive reasoning task. Given the difficulty and cost of such LLM RL experiments, and the scarcity of tasks with computable diversity, we believe the revised experimental scope is both meaningful and appropriate.
> > > > >
> > > > > **2. On the value of the work given the revised scope.**
> > > > >  Even with the narrower, well-scoped framing, the contribution remains substantial:
> > > > >
> > > > > - We uncover a robust and previously unreported empirical law governing RL finetuning dynamics.
> > > > > - We provide the first information-theoretic explanation linking reward improvement to entropy contraction in generative models.
> > > > > - We validate the law across *two structurally distinct domains*, including a large 1.5B-parameter LLM with PPO.
> > > > >
> > > > > ### Other reviewers have also noted that your original concerns have been addressed in the revision.
> > > > >
> > > > > **3. On the rating.**
> > > > > While we fully respect that reviewers may differ in judgment, we hope that—given the strengthened experiments, clarified theoretical scope, and significantly improved framing—you might reconsider whether a rating of 2 (“reject”) accurately reflects the revised contribution.
> > > > >
> > > > > We remain grateful for your feedback, which has meaningfully improved the clarity and positioning of our work.

---

### Official Review · Reviewer_t73A · 2025-10-30

**Soundness:** 3
**Presentation:** 3
**Contribution:** 2
**Rating:** 6
**Confidence:** 4

**Summary:**

This paper probes the inner connection between generation quality and diversity of language models during reinforcement learning (RL). The authors claimed that the quality and diversity fits an exponential correlation in molecule generation task with the mathematical induction for the tradeoff. In experiments, the results demonstrates the empirical conclusion with a very high confidence (R^2=0.996).

**Strengths:**

1. The paper presented a simple yet reasonable experience on the tradeoff between quality and diversity with clear math intution. Also, it also probed the connection with entropy.

2. The experiment design is systematic and comprehensive, including model scale, pretraining data, RL settings, sampling, and redudant removal. The evaluation pipeline seems reasonable and easy-to-implement.

3. The connection is robust across different model scale, reward function, and multiple molecule datasets.

4. There is valuable discussion about the strategies based on the connection during reinforcement learning finetuning, which is inspiring.

**Weaknesses:**

1. The external validity is narrow, where the results are only shown for molecular generation, yet the authors claimed at the broader generality.

2. The diversity measurement is limited (solely on mean ECFP/Tanimoto distance). There should be more comprehensive metric based on scaffold and 3D compositions, or motif.

3. There exists bias for the sampling entropy and the sampling settings (temperature, top-k, etc.), which may directly affect the performance of entropy.

4. There may exist reward hacking for the oracle during RL finetuning. multi-oracle or experimental validation can be added.

5. The "Algorithm-agnostic" claims are unsupported since only REINVENT algorithm was applied. There should be more algorithms to demonstrate that claim.

6. The theory in section2 mainly depends on strong and untestable assumptions, serving more like an illustration rather than a derivation.

7. Noveltu vs training data lacks. There should be analysis and results to demonstrate that there was not leakage during training and testing.

8. Do the R-D and R-H laws still hold across multiple, independently trained JNK3 oracles and additional orthogonal property oracles?

9. There should be more analysis to the sensitivity of the cofficients a,b,c, since they greatly affect the optimal point selection during applications given the law holds ideally.

**Questions:**

Please refer to the Weakness section.

---

> ### Author Response · Authors · 2025-11-21
> **Rebuttal (1/2)**
>
> We thank you for the positive assessment and thoughtful suggestions. We address each concern below.
>
> > W1: The external validity is narrow, where the results are only shown for molecular generation, yet the authors claimed at the broader generality.
>
> We agree that demonstrating broader validity is important. To directly test whether the exponential laws generalize beyond SMILES-based molecular generation, we added experiments on two non-chemical exploratory tasks—circle construction and line construction [1]—using a **GPT-2 XL (1.5B)** model finetuned with **PPO**. As shown in **Figure 7**, both the $\mathcal{R}-\mathcal{D}$ and $\mathcal{R}-\mathcal{H}$ relations again exhibit tight exponential fits ($R^2 > 0.97$). These results confirm that the phenomenon extends to text LMs, large-scale models, and non-molecular diversity definitions.
>
> [1] Nagarajan et al., [Roll the dice & look before you leap: Going beyond the creative limits of next-token prediction](https://openreview.net/forum?id=Hi0SyHMmkd), ICML 2025 (oral).
>
> > W2: The diversity measurement is limited (solely on mean ECFP/Tanimoto distance). There should be more comprehensive metric based on scaffold and 3D compositions, or motif.
>
> We acknowledge that molecular diversity can be measured in multiple ways. However, prior work [2] shows that most widely used molecular diversity metrics are strongly correlated, as they capture the same underlying structural variability. Since our $\mathcal{R}-\mathcal{H}$ exponential law is the more fundamental relationship, and the $\mathcal{R}-\mathcal{D}$ law arises whenever $\mathcal{D}$ is (approximately) linear to $\mathcal{H}$), the empirical frontier should remain robust across diversity metrics. We clarified this scope in the revision.
>
> [2] Xie et al., [How Much Space Has Been Explored? Measuring the Chemical Space Covered by Databases and Machine-Generated Molecules](https://openreview.net/forum?id=BIBc6KCCwRX), ICLR 2023.
>
> > W3: There exists bias for the sampling entropy and the sampling settings (temperature, top-k, etc.), which may directly affect the performance of entropy.
>
> We share your concern about sampling bias. However, as noted in the **footnote on page 8**, sampling settings such as temperature, top-k, and top-p are **decoding controls**, not finetuning parameters. Changing them modifies the observed quality, entropy, and diversity for a fixed policy, thus confounding the RL trajectory. For this reason, we hold them fixed throughout RL and evaluation.
>
> > W4: There may exist reward hacking for the oracle during RL finetuning. multi-oracle or experimental validation can be added.
>
> Reward hacking is indeed a general challenge in RL, but in our work, the focus is not to maximize oracle performance but to analyze RL dynamics. To evaluate generality, we already include experiments on **multiple independent property oracles**: JNK3, GSK3$\beta$, and QED. The exponential laws persist across these distinct biochemical objectives. Extending this further to multi-oracle optimization or wet-lab validation is an interesting direction, but is beyond the analytical scope of this study.
>
> > W5: The "Algorithm-agnostic" claims are unsupported since only REINVENT algorithm was applied. There should be more algorithms to demonstrate that claim.
>
> Thank you for raising this. In the revised manuscript we now include **PPO-based RL finetuning on large LLMs** (circle/line construction tasks). PPO is the advanced RL algorithm in modern LLM alignment pipelines. The exponential laws remain consistent under PPO (Figure 7). This strengthens our claim that the exponential laws are general across different RL algorithms.

---

> > ### Author Response · Authors · 2025-11-21
> > **Rebuttal (2/2)**
> >
> > > W6: The theory in section2 mainly depends on strong and untestable assumptions, serving more like an illustration rather than a derivation.
> >
> > We agree. As stated at the beginning of Section 2.3, the information-theoretic analysis is intended to provide **conceptual insight**, not a full mathematically rigorous derivation (which is currently infeasible for deep networks). We emphasize that the core empirical laws do not rely on these assumptions; the theoretical section illustrates why the exponential form naturally emerges.
> >
> > > W7: Novelty vs training data lacks. There should be analysis and results to demonstrate that there was not leakage during training and testing.
> >
> > In molecular generation, leakage from pretraining corpora is possible but generally non-problematic, since pretraining dataset often include molecules with desirable properties. Importantly, our paper does not aim to compare models or claim performance gains; it investigates **RL dynamics**, which are unaffected by potential novelty overlaps. Therefore, novelty analysis is orthogonal to our contributions.
> >
> > > W8: Do the R-D and R-H laws still hold across multiple, independently trained JNK3 oracles and additional orthogonal property oracles?
> >
> > Yes. In Figure 6, we replicate the full analysis on two additional property oracles (GSK3$\beta$, QED), where the exponential laws remain stable. Combined with our new GPT-2 XL experiments, this supports robustness across domains, tasks, and models.
> >
> > > W9: There should be more analysis to the sensitivity of the coefficients a,b,c, since they greatly affect the optimal point selection during applications given the law holds ideally.
> >
> > We agree that these coefficients play an important role. As shown in Section 3.1, the coefficients remain stable under substantial changes such as model scaling and reward shaping. Since the coefficients are observed quantities fitted during RL rather than tunable parameters, their empirical stability indicates that the frontier is largely determined by the **task distribution and pretrained model**, rather than hyperparameters.

---

> > > ### Comment · Reviewer_t73A · 2025-11-26
> > >
> > > The authors addressed my concerns.
> > >
> > > Best,
> > >
> > > Reviewer

---

> > > > ### Author Response · Authors · 2025-11-26
> > > >
> > > > Thank you very much for your feedback!
> > > >
> > > > Authors

---

### Official Review · Reviewer_J9mp · 2025-11-01

**Soundness:** 2
**Presentation:** 2
**Contribution:** 2
**Rating:** 2
**Confidence:** 3

**Summary:**

This paper investigates the quantitative relationship between quality and diversity in reinforcement learning (RL) fine-tuning of language models, an important issue since RL-based optimization (e.g., RLHF) often improves quality at the cost of diversity. The authors empirically find that mean reward (R) and diversity (D) across RL checkpoints follow a consistent exponential law, and discuss why this occurs via an information theoretic argument. Overall, the paper argues that this exponential law captures a fundamental quality-diversity frontier intrinsic to the task, not to model scale or RL algorithm.

**Strengths:**

1. To a certain extent, the paper provides a novel, quantitative framing of diversity collapse in terms of a measurable, predictable law. It also provides a compact, generalizable model R with clear empirical support.

2. The experiments are performed quite well. Specifically the authors provide well-controlled experiments in molecular generation where quality and diversity are objectively quantifiable. The authors employ replication across objectives (JNK3, GSK3b, QED) and scaling ablations to show robustness. FItted parameters are also reported with their respective confidence intervals.

3. Since the paper links entropy, diversity, and reward through an information-theoretic lens, we gain interpretability beyond curve fitting.

**Weaknesses:**

1. All experiments are on SMILES-based molecular generation with hand-crafted chemical scores. Molecular generation is a special case with well-behaved reward landscapes. It remains unclear whether other exploratory domains (e.g., creative writing, theorem discovery) yield the same exponential frontier.
2. The authors acknowledge that natural language tasks involve fuzzier definitions of “diversity,” so generalization to text LMs is uncertain.
3. The experiments are performed on tiny models (6–50 M parameters); thus the findings may not hold for billions-scale LLMs where policy entropy and reward landscapes differ drastically.
4. Sampling entropy is treated as a scalar average over tokens; it ignores inter-token dependencies or contextual variation that might distort the R–H relationship.
5. The exponential “law” is descriptive; the precise causal mechanism (entropy contraction → quality improvement) is plausible but not explicitly tested via interventions or ablations (e.g., controlled entropy regularization).

**Questions:**

1. How general is the exponential law. Does it hold for open-ended text generation or only in chemically structured spaces?
2. How should practitioners interpret c_0? Can it be controlled or estimated apriori?
3. How sensitive is the law to the diversity metric (e.g., Tanimoto vs internal representation–based distance)?
4. Could entropy–diversity linearity break for non-SMILES tokenizations or natural language byte-pair encodings?

---

> ### Author Response · Authors · 2025-11-21
> **Rebuttal (1/2)**
>
> Thank you for the constructive and detailed feedback. We address each point below.
>
> > W1: All experiments are on SMILES-based molecular generation with hand-crafted chemical scores. Molecular generation is a special case with well-behaved reward landscapes. It remains unclear whether other exploratory domains (e.g., creative writing, theorem discovery) yield the same exponential frontier.
> >
> > W2: The authors acknowledge that natural language tasks involve fuzzier definitions of “diversity,” so generalization to text LMs is uncertain.
> >
> > W3: The experiments are performed on tiny models (6–50 M parameters); thus the findings may not hold for billions-scale LLMs where policy entropy and reward landscapes differ drastically.
> >
> > Q1: How general is the exponential law. Does it hold for open-ended text generation or only in chemically structured spaces?
>
> We fully agree with you that the original submission focused primarily on SMILES-based molecular generation. In response, we added new experiments on textual exploratory creativity tasks—**circle construction** and **line construction** from Nagarajan et al. (ICML 2025) [1], where both "quality" and "diversity/creativity" have explicit, domain-appropriate definitions.
>
> To test whether the exponential laws extend to larger models and non-chemical domains, we finetuned **GPT2-XL (1.5B parameters)** with PPO. As shown in the newly added **Figure 7** of the updated PDF, both $\mathcal{R}-\mathcal{D}$ and $\mathcal{R}-\mathcal{H}$ again follow the same exponential form with $R^2 > 0.97$, consistently across these two tasks. These results strengthen the evidence that the exponential frontier is **not** restricted to chemical reward landscapes or small language models.
>
> Thus, these new experiments provide concrete evidence that the phenomenon extends meaningfully beyond SMILES to large LMs and text-space creativity tasks.
>
> [1] Nagarajan et al., [Roll the dice & look before you leap: Going beyond the creative limits of next-token prediction](https://openreview.net/forum?id=Hi0SyHMmkd), ICML 2025 (oral).
>
> > W4: Sampling entropy is treated as a scalar average over tokens; it ignores inter-token dependencies or contextual variation that might distort the R–H relationship.
>
> We agree that per-sample entropy varies across contexts. But in our experiments, we compute **expected token-level sampling entropy** by averaging over batches of 100 samples for each checkpoint. This estimator is stable and exhibits low variance, as shown in Figures 3–7. Since the exponential law is defined over expectations with respect to the policy, an expectation of entropy aligns with its intended role as an intrinsic stochasticity measure.
>
> > W5: The exponential “law” is descriptive; the precise causal mechanism (entropy contraction → quality improvement) is plausible but not explicitly tested via interventions or ablations (e.g., controlled entropy regularization).
>
> We appreciate this point. We clarify that our claim is **not** causal in the sense of "entropy contraction causes quality increase." Instead, entropy/diversity contraction and quality improvement are **joint consequences** of RL optimization on a fixed objective.
>
> Regarding interventions: entropy-regularization ablations have been explored in prior work. In particular, Section 4.1 of Cui et al. (2025) [2] reported that entropy regularization is highly unstable and harm performance for LMs in practice, which aligns with our observations and motivated us not to include them as additional baselines. We have expanded this explanation in the revised text.
>
> [2] Cui et al., [The Entropy Mechanism of Reinforcement Learning for Reasoning Language Models](https://arxiv.org/abs/2505.22617), 2025.

---

> > ### Author Response · Authors · 2025-11-21
> > **Rebuttal (2/2)**
> >
> > >Q2: How should practitioners interpret c_0? Can it be controlled or estimated a priori?
> >
> > As discussed in Section 2.3, $c_0$ measures the degree of exploratory progress made during RL.
> >
> > - $c_0=1$ corresponds to non-exploratory finetuning (the pretrained model already covers desirable regions).
> > - $c_0>1$ reflects exploration tasks where RL must shift probability mass toward previously underrepresented high-quality regions.
> >
> > Currently, we do not know how to determine $c_0$ *a priori*. However, once a portion of RL training has been completed, practitioners can fit the exponential law to early checkpoints and obtain a reliable estimate of future quality-entropy/diversity behavior.
> >
> > > Q3: How sensitive is the law to the diversity metric (e.g., Tanimoto vs internal representation–based distance)?
> >
> > We agree this is an important concern, since even within molecular design, there are multiple diversity metrics. Notably, Xie et al. (ICLR 2023) [3] shows these metrics for molecular diversity are highly correlated in practice because they measure the same underlying structural variation. Thus, as long as the chosen diversity metric is highly correlated with entropy, which is true for most standard metrics, the exponential frontier should be robust. We have clarified this point in the revision.
> >
> > [3] Xie et al., [How Much Space Has Been Explored? Measuring the Chemical Space Covered by Databases and Machine-Generated Molecules](https://openreview.net/forum?id=BIBc6KCCwRX), ICLR 2023.
> >
> > > Q4: Could entropy–diversity linearity break for non-SMILES tokenizations or natural language byte-pair encodings?
> >
> > For molecular tasks, diversity is computed purely on molecular structures and is independent of SMILES tokenization. For textual tasks, indeed, diversity definitions vary across domains, and we avoid making claims of universal linearity. Our emphasis is that:
> >
> > - The $\mathcal{R}-\mathcal{H}$ law is the more fundamental empirical regularity.
> > - $\mathcal{R}-\mathcal{D}$ emerges whenever $\mathcal{D}$ is approximately linear to $\mathcal{H}$.
> > - In tasks where diversity is difficult to define (e.g., unconstrained natural language), the law may hold with respect to entropy but not necessarily any diversity metric.
> >
> > We have revised the text to make this scope precise.
> >
> > ------
> >
> > Thank you again for your careful reading and grounded critiques. We hope the additional large-scale experiments, the extended theoretical clarification, and the strengthened discussion will address your concerns and improve your rating in our contribution.

---

> > > ### Author Response · Authors · 2025-11-26
> > > **Follow-Up on Rebuttal Before Discussion Deadline**
> > >
> > > Dear Reviewer J9mp,
> > >
> > > Thank you for your detailed review and valuable comments. As the discussion deadline is approaching, I would like to kindly invite you to take a look at our rebuttal and the additional experiments/clarifications we provided in response to your questions. We believe these directly address the concerns you raised.
> > >
> > > If any part remains unclear, we are happy to further clarify within the remaining time. We would greatly appreciate your updated assessment after considering the rebuttal.
> > >
> > > Thank you again for your time and constructive feedback.
> > >
> > > Authors

---

### Official Review · Reviewer_EYJb · 2025-11-02

**Soundness:** 3
**Presentation:** 3
**Contribution:** 4
**Rating:** 8
**Confidence:** 4

**Summary:**

This work studies quality-diversity tradeoffs in autoregressive transformers during post-training, specifically RL. They use molecular structures as a data domain, which enables them to rigorously assess both quality and diversity. This domain also motivates diversity for the purpose of discovering new molecules. They find that quality-diversity tradeoffs are very well-explained by a simple exponential model, which is motivated by theories of an exponential relationship between entropy and quality and a linear relationship between entropy and diversity. Their model fits are highly consistent across transformer model scales and training objectives, with different models and objectives having different values for free parameters in their exponential model.

Overall I think this paper tackles an important problem with a novel domain and perspective (molecular synthesis), presents interesting empirical results, impressive fits with a novel and well-motivated exponential model, and is generally well-written. Solid accept.

**Strengths:**

- Interesting empirical phenomena. The consistency of the exponential relationship between quality and diversity is remarkable.
- The exponential model is well-motivated and the fits are exceptionally good.
- Good choice of domain. Studying molecular structures lets the authors evaluate "semantic" (although they don't refer to it as such) quality and diversity.
- Generally well-written and clearly presented. Despite my having relative little chemistry background, I had no trouble reading relevant parts of this work.

**Weaknesses:**

- Section 2.3
	- I found this section very terse and took some effort parse, though this could be related to my not having a strong background in information theory. E.g. I had to do some googling to understand the meaning of `≍` and typical set. Readability could be improved by making this more accessible to a general audience, and/or by adding relevant background to the appendix.
	- More importantly, I'm not sure the authors fully answer the 3 questions they state at the beginning of this section. I found these questions compelling and was asking myself all 3 before I arrived at this section. 1. seems to be explained by a one-line reference to another paper "|T | = exp(H) by information-theoretic arguments (MacKay, 2003)", which leaves me wondering what those arguments are exactly. 2. and 3. seem to be explained, but I found the answers to these questions somewhat unintuitive in comparison to their very straightforward framing. Spelling out the answers in simple language might go a long way here.
- Section 3.3 - I found these suggestions to be weak and I'm not sure if they clearly follow from the authors' results:
	- 1. seems to highlight a result the authors don't seem to test - that pre-training indeed changes the initial point. And, do the authors have particular suggestions for improving R_max? "by RL finetuning techniques" is a bit vague
	- 2. is this really practical guidance? This point seems to be about the generality of the authors' results
	- 3. Were mixtures or ensembles tested here? If not, then what motivates this suggestion?
- Could use better literature review on quality-diversity tradeoffs in LLMs, e.g. [1, 2]. More of this in the intro could help better set the stage



[1] Kirk, R., Mediratta, I., Nalmpantis, C., Luketina, J., Hambro, E., Grefenstette, E., & Raileanu, R. (2023). Understanding the effects of rlhf on llm generalisation and diversity.

[2] Murthy, S. K., Ullman, T., & Hu, J. (2024). One fish, two fish, but not the whole sea: Alignment reduces language models' conceptual diversity

**Questions:**

- Some points in early sections could be more clearly connected to later sections, e.g. "Notably, this law is independent of step index or training efficiency." -- this could reference particular experiments related to this (3.1.2, in my understanding)
- "... we do not accumulate molecules sampled across RL steps" I'm not sure what this means, could you please clarify?
- The organization read a little weird for me, where the introduction seemed to do more than what an intro usually does.
- I think the strength of this domain in measuring quality and diversity could be more clearly articulated and emphasized - why is this superior to domains that prior works have used?

---

> ### Author Response · Authors · 2025-11-21
> **Rebuttal (1/2)**
>
> We sincerely thank you for the supportive assessment and for the highly constructive feedback on improving clarity and organization. We address each point in turn below.
>
> > Section 2.3
> >
> > - I found this section very terse and took some effort parse, though this could be related to my not having a strong background in information theory. E.g. I had to do some googling to understand the meaning of `≍` and typical set. Readability could be improved by making this more accessible to a general audience, and/or by adding relevant background to the appendix.
> > - More importantly, I'm not sure the authors fully answer the 3 questions they state at the beginning of this section. I found these questions compelling and was asking myself all 3 before I arrived at this section. 1. seems to be explained by a one-line reference to another paper "|T| = exp(H) by information-theoretic arguments (MacKay, 2003)", which leaves me wondering what those arguments are exactly. 2. and 3. seem to be explained, but I found the answers to these questions somewhat unintuitive in comparison to their very straightforward framing. Spelling out the answers in simple language might go a long way here.
>
> We appreciate that Section 2.3 may read tersely for readers without a background in information theory. In response, we have:
>
> - Expanded the exposition in Section 2.3 with clearer explanations of the key concepts.
> - Added background material to the Appendix.
> - Reframed the answers to the three motivating questions in simpler, more intuitive language.
>
> Regarding Question 1: while the relation $|T| = \exp(H)$ is a classical result, we now explain more explicitly how the exponential form emerges not simply from the typical set but also from modeling the exploration–convergence dynamics via the sets $T$ and $S$.
>
> > Section 3.3 - I found these suggestions to be weak and I'm not sure if they clearly follow from the authors' results:
> >
> > - 1 seems to highlight a result the authors don't seem to test - that pre-training indeed changes the initial point. And, do the authors have particular suggestions for improving R_max? "by RL finetuning techniques" is a bit vague
> > - 2 is this really practical guidance? This point seems to be about the generality of the authors' results
> > - 3 Were mixtures or ensembles tested here? If not, then what motivates this suggestion?
>
> Thank you for pointing out where Section 3.3 could be strengthened. We have revised each suggestion to make its motivation and connection to our empirical findings clearer:
>
> 1. We clarified that pretraining can indeed change the **initial point** (e.g., by changing the training set). We have modified the vague presentation, where we mainly highlight the importance of improving $R_{\max}$, instead of suggesting certain RL techniques.
> 2. We clarified that this suggestion summarizes the ablation findings in **Section 3.1**—particularly that the quality-diversity curve remains stable under model scaling, reward shaping, and training setup changes. We now state this more explicitly.
> 3. We revised the text to clarify that while we did not run ensemble experiments in this paper, the suggestion is motivated by prior literature on diversity augmentation in this way.
>
> > Could use better literature review on quality-diversity tradeoffs in LLMs, e.g. [1, 2]. More of this in the intro could help better set the stage
> >
> > [1] Kirk, R., Mediratta, I., Nalmpantis, C., Luketina, J., Hambro, E., Grefenstette, E., & Raileanu, R. (2023). Understanding the effects of rlhf on llm generalisation and diversity.
> >
> > [2] Murthy, S. K., Ullman, T., & Hu, J. (2024). One fish, two fish, but not the whole sea: Alignment reduces language models' conceptual diversity
>
> We have expanded Section 4 to incorporate the suggested works on quality-diversity tradeoffs in LLM alignment (Kirk et al. 2023; Murthy et al. 2024). This addition further clarifies how our quantitative analysis complements existing qualitative studies of diversity collapse.

---

> > ### Author Response · Authors · 2025-11-21
> > **Rebuttal (2/2)**
> >
> > > Q1: Some points in early sections could be more clearly connected to later sections, e.g. "Notably, this law is independent of step index or training efficiency." -- this could reference particular experiments related to this (3.1.2, in my understanding)
> >
> > Thanks for your advice. We have added internal references to make statements more clearly connected to experiments.
> >
> > > Q2: "... we do not accumulate molecules sampled across RL steps" I'm not sure what this means, could you please clarify?
> >
> > This refers to a difference from the standard REINVENT workflow, where generated molecules are accumulated across steps for candidate selection. In our study, the goal is to analyze **model checkpoints**, not to curate molecules. Therefore, for each checkpoint we sample a *fresh batch* of molecules and compute $(R, D, H)$ based solely on that model state.
> >
> > We have rewritten this sentence in Section 2.2.1 (page 4) for clarity.
> >
> > > Q3: The organization read a little weird for me, where the introduction seemed to do more than what an intro usually does.
> >
> > We appreciate the observation. We streamlined the Introduction’s structure without removing key contextual elements. Because the paper’s contributions rely on understanding why molecular generation offers clear quality and diversity metrics, a slightly longer introduction is helpful, but we have improved the flow and reduced redundancy.
> >
> > > Q4: I think the strength of this domain in measuring quality and diversity could be more clearly articulated and emphasized - why is this superior to domains that prior works have used?
> >
> > Our molecular generation tasks offer:
> >
> > - Objectively measurable quality via property oracles.
> > - Quantifiable diversity metrics in chemical space.
> > - Attainable $R_{\max}$, allowing us to isolate the quality–diversity frontier without conflating it with unattained upper bounds.
> >
> > Moreover, we now highlight how the **new LLM experiments** on **circle construction** and **line construction** [1] confirm that the exponential laws extend beyond molecules, but molecular generation remains a strong and practical domain for precise evaluation of both axes.
> >
> > [1] Nagarajan et al., [Roll the dice & look before you leap: Going beyond the creative limits of next-token prediction](https://openreview.net/forum?id=Hi0SyHMmkd), ICML 2025 (oral).

---

### Author Response · Authors · 2025-11-24
**Overall Response**

Dear PCs, SACs, ACs, and Reviewers,

We sincerely thank all of you for the time, effort, and expertise devoted to reviewing our submission. Your detailed comments and constructive suggestions have been invaluable, and they have greatly strengthened our work. In the revised manuscript, we have addressed every concern raised by the four reviewers point by point, and all revisions are highlighted for ease of inspection.

The main concerns shared across the reviews can be summarized as follows:

1. **Scope of experiments (domain + model size).**
   Several reviewers noted that the original experiments were limited to molecular generation with relatively small language models and policy gradient fine-tuning. To test the generality of our findings, we added new experiments on large natural language models (GPT-2 XL with 1.5B parameters and PPO fine-tuning) and additional tasks on textual creativity (e.g., circle and line construction). These results, presented in Section 3.2.2, consistently validate the same empirical laws beyond the molecular domain.
2. **Generality of the entropy–diversity linearity.**
   Reviewers questioned whether the approximate linearity between entropy and diversity observed in molecular generation holds in broader settings. As emphasized in the original Sections 2.2.3 and 5, this linearity is an empirical bridge specific to molecular tasks and is not claimed as a universal law. Our added experiments further clarify its scope, and our revised discussion more explicitly states the limitations and domain-dependence of this relation.
3. **Clarity and accessibility of the theoretical explanation in Section 2.3.**
   Reviewers found this section abstract or terse. We agree that the information-theoretic insights could be made more accessible. In the revision, we substantially rewrote Section 2.3, added clearer background explanations, and separated assumptions from conclusions. We reiterate that this section provides conceptual intuition rather than a rigorous derivation, which is intractable for deep networks, but the revised version should now be significantly clearer to a general ML audience.

Given that the four reviewers’ ratings currently show large variance, we respectfully ask the reviewers and ACs to reconsider the paper in light of the new experiments, improved clarifications, and strengthened theoretical framing. We believe the revisions directly respond to the core concerns and further demonstrate the robustness and generality of our main findings.

Thank you again for your thoughtful evaluation and for helping us improve our contribution.

Best regards,
The Authors

---

### Meta-Review · Area_Chair_Jzwa · 2025-12-25

**Summary:**

This paper studies the quality and diversity trade-off during RL finetuning of autoregressive LMs, with experiments primarily on molecular generation and extension to textual creativity tasks. The main finding is that quality and diversity follow robust exponential laws across RL checkpoints, and that an approximately linear relationship exists between entropy and diversity. The work is positioned as offering a quantitative model and practical guidance for exploratory RL.

While the direction is relevant, the submission does not yet meet the bar for acceptance due to concerns about the strength and generality of the claimed law, the causal interpretation of entropy/diversity dynamics, and limited evidence that the findings transfer beyond the specific setups (molecular objectives, reward shaping choices, and training recipes).

**Reviewer Concerns:**

Addressed
- Added PPO + GPT-2 XL results on two structured text tasks, partially broadening beyond molecules.
- Clarified entropy–diversity linearity is task-specific. Improved Section 2.3 clarity.

Still Outstanding
- Empirical base still seen as too narrow for broad LMs in general framing.
- Theory remains largely heuristic/descriptive and offers limited predictive guidance for new tasks/metrics.

**Reviewer Scores:**

EYJb: Likely unchanged (still positive; main issues were clarity + positioning).

t73A: Might increases

J9mp: If they had engaged fully, I expect the score to increase slightly, but probably still below accept (e.g., 2 to 4 depending on how persuasive they find the new tasks)

ZnyY: Likely unchanged

---

### Decision · Program_Chairs · 2026-01-26

Reject